# A Christian Pluralistic Hypothesis: To Bonhoeffer and beyond—A World Christianity

Greg Gorsuch 

Independent Researcher, Seattle, WA 98020, USA; greggorsuch@fuller.edu

**Abstract:** This essay investigates Bonhoeffer's undeveloped concept of "Unconscious Christianity" and how a protracted understanding of his religionless Christianity in a culture "come of age" points to a viable Christian pluralistic hypothesis—how Christ and the Spirit are redemptively active outside the church. Bonhoeffer's living faith action transcends his theology, revealing unconscious dynamics within our interactions that reveal antecedent relational dynamics that open to the redemptive process, which transcend but do not obviate the cognitive elements of faith. New scriptural themes and concepts of relationality and perichoretic metaphysics bring greater biblical coherence and meaning to one particular biblical passage that has apparently remained meaningless (Matt 12:32). This new meaning and coherence within the scriptures radically alter Christianity's relationship to the outside world and transforms our understanding of the Great Commission.

**Keywords:** Christian pluralistic hypothesis; Matthew 12:32; Bonhoeffer; *perichoresis*; *analogia spiritus*; trinitarian metaphysics; relational metaphysics



Aslan: "Child [Emesh], all the service thou hast done to Tash, I account as service done to me. ... I take to me the services which thou hast done to him. ... Therefore if any man swear by Tash and keep his oath for the oath's sake, it is I who reward, ... unless thy desire had been for me thou wouldst not have sought so long and so truly. For all find what they truly seek". (Lewis 1956, p. 202).

It is to the prodigals—to those who exhaust all their strength in pursuing what seems to them good and who, after their strength has failed, go on impotently desiring—that the memory of their Father's house comes back. If the son had lived economically he would never have thought of returning. (Weil 1956, p. 211).

## 1. Introduction

This essay represents a pneumological undertaking and paradigmatic shift in considering a developing understanding of *perichoresis* as a metaphysical shift from *analogia entis* or *analogia relationis* to *analogia spiritus*. Such novel orientations allow us to offer a viable Christian pluralistic hypothesis theologically and biblically. The thesis takes root in Bonhoeffer's interdisciplinary consideration of social theory that begins with *Sanctorum Communio* and culminates in his undeveloped concept of "unconscious Christian" in his prison letters and papers, similar to Rahner's "anonymous Christian". From this trajectory, we will explore the dynamics of *perichoresis* as a theological and biblical precedent for how the redemptive work of Christ and the Spirit might be active in and outside the church. Unlike the recent work of those like Amos Yong, who are beneficially working at other levels in understanding the work on the Spirit, this thesis seeks to understand the activity of spirit at a more metaphysical level within human and divine relations—one that reveals its power by bringing more significant meaning to heretofore enigmatic Scriptures.

## 2. Bonhoeffer's Lived Faith and Unconscious Christianity

Dietrich Bonhoeffer was a confirmed pacifist, admired Mahatma Gandhi, and even made plans to visit India to meet with him, which never materialized (Zimmermann

2019, p. 3). He maintained his pacifist convictions both before and, surprisingly, after his authorizing the failed assassination of Adolf Hitler, which led to his imprisonment and eventual execution. In an act of lived religious faith of Abrahamic proportions, he believed God led him beyond his penultimate faith convictions (*fides reflexa*), theology, morals (pacificism), and reason via a transcendent faith (*fides directa*) similar to Kierkegaard's "teleological suspension of the ethical". He also questioned the genuineness of the Christian faith in the German Lutheran Church leadership, the vast majority of German "Christians" supporting the Third Reich, and even those remaining compliantly silent (McLaughlin 2020, pp. 155–56; Bonhoeffer 2005, pp. 139–44). Furthermore, he was beginning to consider that non-professing Christians who signed the assassination authorization or identified and resisted the evil within the Third Reich were nevertheless directly acting with Christ as "unconscious Christians" (McLaughlin 2020, p. 157). Being a psychiatrist's son might have sensitized Bonhoeffer to the unconscious dynamics of human action within faith experiences, especially as he tried to make sense of his troubled socio-political world.

During his time in prison, the seeds of a Christian pluralistic hypothesis began to grow. His use of concepts like "unconscious Christianity", "religionless Christianity", and "the world come of age" marked the beginning of a significant theological deviation from his past in search of social and relational criteria that might expose the redemptive activity of the Spirit and Christ beyond the cultic convictions and activity of the church. In a letter to his friend, Eberhart Bethge, he shorthanded three referenced passages that indicate unconscious expressions of the Christian faith in an outline for a future book (Bonhoeffer 2009, p. 491): (1) "Left hand doesn't know what the right hand is doing", referring to Matt 6:3, (2) "Mt 25", referring to the unconscious action within our faith and knowing experiences, as well as the cognitive confusion this creates for many learning about their unconscious actions (Matt 25: 31–46), (3) "Not knowing what to pray", which refers to Rom 8:26b, "for we do not know how to pray as we ought, but that the very Spirit intercedes with sighs too deep for words". All these indicate potential unconscious action within faith experiences. The actions that expose unconscious Christians for Bonhoeffer are "goodness" reflected in a person's actions and a person's Christlike "being for others" who are suffering. These thoughts were embryonic, remaining undeveloped due to his premature death.

### 3. Bonhoeffer and Spirit: The Pulsation between Time and Eternity within Our Relationships

We must consider Bonhoeffer's theological context and position before fully understanding his thoughts on a viable Christian pluralistic hypothesis. He was quite drawn to Karl Barth and the emerging neo-orthodoxy as Barth attempted to correct the abuses of Hegel's liberal Protestantism and concept of Absolute Spirit. Contrary to Hegel, Barth and Bonhoeffer argued that the divine and human spirit were distinct. Nevertheless, Barth maintained a strong Christological emphasis over that of the Spirit of God. Toward the end of his life, however, Barth conceded to Schleiermacher the following:

> What I have already intimated here and there to good friends, would be the possibility of a theology of the third article, in other words, a theology predominantly and decisively of the Holy Spirit. Everything which needs to be said, considered and believed about God the Father and God the Son in an understanding of the first and second articles might be shown and illuminated in its foundations through God the Holy Spirit. (Barth 1982, p. 278)

He refrained from doing so because he thought it was "still too difficult to distinguish between God's Spirit and man's spirit". Compounding the problem was the general continental portmanteau of crashing the original Greek concepts of νοῦς and πνεῦμα into a single word (e.g., *geist*, *esprit*, and *geest*). Furthermore, though Barth was unwilling to give such quarter to liberalism in developing a more robust pneumatology, which led to his ardent Christology, Bonhoeffer's interdisciplinary sensitivity to emerging disciplines of study that were beginning to further acknowledge analogies between divine and human spirit emboldened the younger Bonhoeffer to tread where Barth would not. His doctoral

thesis, *Sanctorum Communio*, investigates the relationship between the divine and human spirit within a Christian social theory. Bonhoeffer was sensing what D B Dabney would later argue—that the word constructs upon a *preceding* breath (spirit):

> We must insist against Barth that it is the *Spirit* of God and not simply the *Word* of God that is properly basic to Christian theology, then against Schleiermacher we must maintain that it is the Spirit *of* God and not *human* spirituality that is the proper subject matter for an appropriate prolegomenon to theology. . . . The Spirit of God is not human spirit aspiring to the divine, but neither is it the subjectivity of God making an object of the human. . . . Rather than *sub*jective or *ob*jective, the Spirit is better conceived as *trans*jective; that is to say, that by which we as individuals are transcended, engaged, oriented beyond ourselves, and related to God and neighbor from the very beginning. (Dabney 1996, pp. 160–61)

Dabney's consideration suggests that a metaphysical grounding (contingency) for faith and theology would better rest in a trinitarian coupling of Christ *and* the Spirit, acknowledging the asymmetrical priority of God's Spirit (breath of God). The revisioning Bonhoeffer in prison was beginning to have similar considerations. This also parallels the convictions of Kierkegaard that humanity is a relationship between the temporal and the eternal that creates our social mutuality as a positive third term while maintaining and intensifying the polarities (Loder and Neidhardt 1992, p. 209):

> Man is a [relation between two factors] . . . the temporal and the eternal. . . . In a relation between two, the relation is the third term as a negative unity. . . . If on the contrary the relation relates itself to its own self, the relation is then the positive third term. . . . If this relation which relates itself to its own self is constituted by another, the relation doubtless is the third term, but this relation (the third term) is in turn a relation relating itself to that which constituted the whole relation. (Kierkegaard 1941, p. 146)

The emergent third term of the relationship is spirit and becomes self or social relations. Bonhoeffer also reflects on this same dynamic:

> Two wills encountering one another form a structure. A third person joining them sees not just one person connected to the other; rather, the will of the structure, as a third factor, resists the newcomer with a resistance not identical with the wills of the two individuals. Sometimes this is even more powerful than that of either individual—or than the sum of all the individuals, if this is at all conceivable. Precisely this structure is objective spirit. (Bonhoeffer 1998, p. 98)

For Bonhoeffer, "objective spirit" is the emergent social structure of community, a living, tangible social relationality. Reflecting the shape of the Trinity, human developmentalist and theologian James Loder is further convinced that "the reality of mutuality becomes self-conscious, or aware of itself as such" (Loder and Neidhardt 1992, p. 291).

The peculiar uniqueness and necessity embodied within the concept of the Trinity is an incarnational imperative of mutuality (IIM)—a divine incarnation that is in and part of the world, not outside it. The IIM maintains that true *filial* intimacy between humanity and God is only possible if an aspect of God becomes human and an aspect of humanity becomes divine. This happens in the Incarnation of God in Christ (bringing the temporal into the godhead) and the complementarity of the Spirit of God (infinite totality) coming into a holistic relationship with the human spirit—the small infinity (bringing the eternal into a *co-conditioning* relationship with human spirit, for eternity transcends cause-and-effect temporality). Zimmermann presents Bonhoeffer's paralleling dynamics in ethics as "the free act of the human will in '*immediate* relationship with God's will,' a relation to be established anew for each concrete occasion . . . 'in the distressful decision of the moment. . . . Only through the depths of our earth and the storms of human conscience does an eternal perspective open up to us'" (Zimmermann 2019, p. 8).[1] In maintaining Kierkegaard's "infinite qualitative difference", such a relationality can only occur through an existential pulsing (tension) between the systole of the eternal and the diastole of the

temporal. Without this, God and humanity cannot have a genuine filial relationship. *Both* time and eternity potentially condition humanity through this cycling from within, not from outside, our interactions. We have all experienced this phenomenon of the two conditioning aspects of the eternal/spirit and time/word every time we are convicted of sin. Loder gives an example of the asymmetric priority of Spirit:

> To be convicted of sin . . . would be unbearable . . . if it were not that such a realization is preceded by the grace that makes such a realization not only bearable but profoundly generative of a new being. The Creator Spirit must create *ex nihilo* in individuals as it created at the beginning of the whole of creation. Thus, mortification precedes the illumination in our accounting of it, but in the sphere of the spirit, the illumination precedes and anticipates the mortification. (Loder 1998, p. 116)

The grace and righteousness of Christ are 'evident only in the sphere of the Spirit'; therefore, pre-conscious *analogia spiritus* must engage before regenerative gestalt. Similarly, Bonhoeffer affirms such dynamics in quoting Augustine, "'You would not seek me if you had not already found me'" (Zimmermann 2019, p. 249). Furthermore, in Bonhoeffer's words, "The person ever and again arises and passes away in time. The person does not exist timelessly; a person is not static, but dynamic. . . . The person is re-created again and again in the perpetual flux of life" (Bonhoeffer 1998, p. 48). Bonhoeffer's sensitivity to these subtle, often unacknowledged dynamics within social relations, eventually opens the door for his later considerations of unconscious actions within faith and provides grounds for a Christian pluralistic hypothesis. Likewise, Bonhoeffer's willingness to fully opening to the Other *and* differentiate from the majority in full autonomy creates his eventual political stand for which he would literally have to pick up his cross.

## 4. A Viable Christian Pluralistic Hypothesis

Today, we live in a much smaller world than Bonhoeffer's because of advances in travel, communication, geo-economics, and extreme social migration. The world is becoming a smaller community daily, and virtually all religions and ethnicities are becoming neighbors, attending the same schools, and working side by side. To avoid exasperating its relationship with the world, the church must theologically finish Bonhoeffer's concerns for developing a lived understanding of how Christ and the Spirit might be redemptively active outside of Christianity. Such a theological undertaking would allow the church to better discern its faith development inside the church and identify the possibility of it outside the church. Customarily, we believe Christ's redemptive action is at odds with, constantly subverting, and transforming the social dynamics of the world. Still, the scriptures also teach that all creation is held together by the creating activity of Christ and the Trinity (Col 1) and, therefore, capable of exhibiting Christ's and the Spirit's redemptive patterns everywhere in the world. For Bonhoeffer, such redemptive dynamics ultimately have no cultic exclusivity even though, for him, they are uniquely and only perfectly found within the Trinity. For "God . . . [is] not at the boundaries but at the center, not in weakness but in strength; . . . God's "beyond" is not what is beyond our cognition! God is the beyond in the midst of our life. . . . The church stands . . . in the center of the village" (Bonhoeffer 2009, pp. 366–67). He finds God powerfully evident in the midst of his sociology, psychology, or under every rock he overturns. Yet, he maintains the uniqueness of Christ's divinity and that all who come to the Father come through Christ. Nevertheless, with Kierkegaard, he was beginning to believe that the *how* of faith was somehow asymmetrically prior to and bore more of the revelational and redemptive weight than the *what* of faith, even though both inextricably connect.

Bonhoeffer's redemptive criteria demonstrating Christian faith, "goodness" and "being for others", express two of the characteristics of love. Though he certainly would have developed these further, alone they remain theologically unfinished and problematic. Goodness alone flounders in our postmodern world as a significant theological marker for redemption. When God suggests that after all of David's life decisions, Jacob's manner of

attaining the blessing, and Job's caustic addressing of God, that these men's hearts were indeed after God all along; goodness becomes a problematic criterion to theologically and socially implement. Likewise, being for others who are suffering eventually falls short of the full wealth of what Paul expresses in love, for he begins by saying, "though I bestow all my goods to feed the poor, and though I give my body to be burned, *but have not love*, it profits me nothing (1 Cor 13:3, my emphasis)". He is suggesting that "being for others" can happen without love. Love is much more complex than typically acknowledged; it also "rejoices in the truth, bears all things, believes all things [resists denial], . . . and endures all things." All of which expand our givenness, not just to the other, but to the knowing event and relationship as a whole (spirit) through which we encounter the other. This leads to the unfinished theological project: what is love?

If God is love, we must critically consider the notion of *perichoresis*—the internal relationship of the Trinity within a social understanding of the Trinity—as its highest expression. In this respect, the human reflection of *perichoresis* becomes a limited and analogical reflection of that ultimate relationality through which we might relate to others and the world. Though it is an emerging dynamic that we experience only in part between us, it is a relationality that is still largely beyond us and into which we are being transformed from one degree to another (2 Cor 3:18). Any behavioral criterion evidencing the Christian faith must begin with a critical understanding of this dynamic. Technically, *perichoresis* exhibits an irreducible relationality that both mediates our *mutuality* while maintaining and intensifying our *personhood* (individuality). In an undistorted relationship to God's spirit, human spirit reflects this divine nature and relationality: "Subjective spirit becomes eternally significant only *in relation to* the absolute spirit" (Bonhoeffer 1998, p. 48, my emphasis). Such a call to relationship invites us to first fully open ourselves to the Other (other persons, God, the world, and even the self, i.e., *holistically* entering our relations) *and* then back into differentiated personhood—for the Trinity never collapses into sheer oneness or Hegelian synthesis. A critical study of Bonhoeffer's dissertation, *Sanctorum Communio*, a Christian social theory, reveals his sensitivity to these perichoretic dynamics even though he never uses the term.[2] In this respect, Bonhoeffer defines being, not as substance, but ultimately in relational terms (Zimmermann 2019, p. 192), which point toward the metaphysics of *perichoresis* and a more profound logic of the spirit as the unconscious dynamics within Christian redemption.

The current issues of pluralism become problematic when we fail to acknowledge these more subtle dynamics within the faith experience and the scriptures (e.g., Bonhoeffer's reference to Matt 25). Such subtleties help to parse genuine from disingenuous aspects of faith, and more problematic for the church, expose potential genuine faith outside the church. Ecclesial confessions and practices defining belief and catechism are needed to center a spiritual community. However, the possibility of genuine Christian faith occurring unconsciously outside the church would have been somewhat disorienting to the early church. The early church's lack of social development may have left it ill-equipped to speculatively cope with what is far more discernable today.

On the other hand, if we can redemptively encounter Christ unconsciously, what is the value of consciously knowing the person and story of Christ? This is similar to intuiting something before it consciously emerges in contrast to the knowledge of that thing after one becomes more conscious of it. For example, every scientific discovery starts with an intuition that something is there—a *holistic* initiatory encounter with a new aspect of reality. The scientist *experiences* something of that reality, then slowly generates identifiable patterns emerging from it, and finally, it bursts forth in a more detailed and functional gestalt. With this conscious encounter comes greater understanding and intimacy through which we engage and know the world, the point being that it is the *openness* (weakness) of the scientist that precipitates reality being able to affect the scientist. It is no different with our personal encounter with Christ. Therefore, nurturing openness between the church and the world might be more important than correctly communicating its (necessary) creeds and

theological dogma. Though cultic practices help define the identity of the Christian faith and unify its members, an overarching exclusivism does not, or does so only pathologically.

Some traditionally difficult passages and scriptural themes present these relational dynamics that provide a viable pluralistic hypothesis. Such passages should caution against sectarian exclusivity and relax any unconscious imperialistic expectations and overdetermining of relations between Christianity and those outside the church. If Christ is salvifically active in the lives of others outside Christianity, then Christians would do better to consider the Great Commission as invitation to consciously knowing Christ while relaxing (not negating) the necessity of a conscious confession of him for salvation. If Christian theology could reveal how Christ's redemption might ultimately span outside Christianity yet maintain that consciously knowing Christ offers greater fullness of life than otherwise, it would transform Christianity's relationship to the outside world.

Yes, Christianity and the scriptures boldly state that Jesus Christ is the unique and exclusive Son of God through whom all humanity must come to know God fully; however, Christ himself noted that *how* he is known takes antecedent precedence (asymmetrical priority) over *what* or who is known. Christianity claims that knowing Christ is necessary for salvation, but powerful scriptural precedent clearly indicates that such a knowing is often unconscious. This, however, does not detract from the Christian imperative that Christ be known; it simply expands how this happens and how Christ must be known while considering each person's contextual capacity to know. These passages also state that *how* Christ is known has everything to do with whether he actually is known, therefore adding a preceding relational disposition or contingency to that knowing. Without this deeper understanding of how Christ is active in the world, Christianity remains dysfunctional with cognitive imperatives rendering its witness to the truth inappropriately imperialistic.

## 5. Extending Bonhoeffer's Exegesis beyond Bonhoeffer

Christian theology attempts to read and employ the whole scriptural witness to its fullest coherence and meaning. To the degree we abandon whole passages and themes in the scriptures because of their theological inhospitableness or apparent incoherence, we live with a dysfunctional theology.

Few passages in the New Testament speak of the resurrected, and even fewer indicate an unconscious criteria indicating their redemption. In Matt 25:31–46, many of the resurrected seem confused as Christ states to each person whether they lovingly related to him or not. There will be many who thought they never encountered him but did, and those who thought they did but did not. "Inasmuch as you did it to one of the least of these My brethren, you did it to me. . . . Inasmuch as you did not do it to one of the least of these, you did not do it to Me". Only those who never practiced Christianity or never consciously believed in Christ would be confused by being identified as having related in some way meaningfully to him, even if through others. Moreover, only those who thought themselves Christ believers would be confused by Christ identifying them as never having loved him, even if through others. If they were loving and genuinely open to knowing "the least of these", then it appears Christ identifies this as loving him. There seems to be a subtle, even hidden, relational criterion at play here that initiates the redemptive process and genuine faith.

Matthew 25 is further clarified when Jesus reveals in Matt 7:21–23 that "Not everyone who says to Me, 'Lord, Lord,' shall enter the kingdom of heaven. . . . Many will say to Me in that day, 'Lord, Lord, have we not prophesied in Your name, . . . and done many wonders in Your name?' And then I will declare to them, I never knew you; depart from Me!" Note that Christ does not say a few or some 'Christians', but *many*, and not that he once knew them and later he did not; he *never* knew them. Therefore, Christ's constant call to believe and follow him carries the soteriological criterion that such belief necessitates *authentic* relationship and a *genuine* knowing of him. Just like beliefs, knowing a person is conditioned by *how* we relate as much as to whom or what we relate.

Even more shocking, Jesus requires this antecedent *how* of belief over the cognitive aspect of belief in him! This presents a more precise idea of the *necessary* preceding activity— a relational disposition. After the Pharisees misidentified him as working with Satan, Christ, knowing their theological sophistication dissects their accusation, revealing their unconscious dispositional mistake and presuppositional closure to knowing his complete identity. *He resituates salvation from what is thought about him to* how *they came to those thoughts*. "Anyone who speaks a word against the Son of Man, *it will be forgiven him*; but whoever speaks against the Holy Spirit, it will not be forgiven him" (Matt 12:32, my emphasis). To know Christ as God (1 Cor 2), one must allow the wholeness of Christ (God)—the Holy Spirit—to affect *their whole self*—their spirit—in order for that relationship to be genuine. Only in such encounters can our knowledge and understanding transform to new states of knowing Christ *in his divinity*, knowing the truth of others and ourselves, and knowing the genuine truth of anything (1 Cor 8:1–2). This parallels his insistence that those who want to know him must pick up their cross daily (Matt 10:37–39; Luke 14:26,27,33) and become spirit, exposing themselves to the Spirit of God (2 Cor 3:16–18). To the degree a person bifurcates into multiple selves, harboring untouchable beliefs hidden from conditioning by the truth of the Other (the world), they cannot truly know the Other or the world—they imprison the self within self. We must constantly allow our knowledge, beliefs, and actions to expand into greater fullness of life, which often necessitates re-paradigming. With this, Paul indicates that transformation into the image of Christ is an ongoing series of transformations for every true Christian throughout their lifetime, for the church through history, and the world through time—from one degree to another (2 Cor 3:18; 1 Cor 15:20–28).

Jeremiah says a person will seek God and find God when they search for God with *all* their heart (Jer 29:13), not a portion of their heart, not with a bifurcated soul in which they suppress various aspects of self. When the heart wills one thing (passion), the person becomes spirit—they are whole.[3] The Pharisees' undertow of professional and religious status, theological acumen, employment security, power, and privilege was not allowing them into the swirling transformative matrix of the forming relational unity wherein they could fully experience and identify Christ within his fullness—the Spirit of Christ (of God). When participants consciously or subconsciously withhold themselves or part of themselves from the emerging matrix of the relationship (not the other, but that which mediates the relationship), they, in their freedom, bind the two hands of the Irenaean God (Christ and the Spirit). Within Christianity, the disposition of closedness is sin, not the antiquated failure to maintain established laws and moral codes.

Kierkegaard's Johannes Climacus insists that to encounter the Other genuinely, to be transformed, "it is necessary to risk everything". Encounter with a living God necessitates vulnerably opening all of oneself to the unknown. Only in entering our weakness in humility can the Spirit relate God's fullness to our fullness. Kierkegaard hints at the *how* of belief in a short parable:

> If one who lives in the midst of Christendom goes up to the house of God, the house of the true God, with the true conception of God in his knowledge, and prays, but prays in a false spirit; and one who lives in an idolatrous community prays with the *entire passion of the infinite*, although his eyes rest upon the image of an idol: where is there most truth? The one prays in truth to God though he worships an idol; the other prays falsely to the true God, and hence worships in fact an idol. (Kierkegaard 1941, p. 179, my emphasis)

Likewise, in neo-Platonic language, Noel O'Donoghue similarly notes the following:

> Sharing then, participation (in the active sense) is, at the source, at once the sharing of infinite sharing, and the giving of an infinite capacity for receiving: infinities meet in the finite. The creature is no less infinite than the creator, in the infinity of its radical dependence, its radical nothingness: on this ground rests the infinity of its receptivity: *homo capax Dei*. The mystic makes his own of this negative immensity of openness to the infinite that shares its own being, and in

> this lived appropriation, *experiences the logic of infinity*, experiences that finitude reaching to the infinite, which is the centre of all creativity as it is the centre of all prayer. (O'Donoghue 1979, p. 177, my emphasis)

Spirit, both divine and human, is our infinity, all that we are. Only when the Spirit of God guides us into all truth (John 16:7–15) and engages our whole self (our spirit) can we truly know Christ in the dynamic exchange of *analogia spiritus*. This is the necessary state of openness (the cross) Jesus requires for being known as the Christ, as God (Matt 12:32). Unless we release all that we are into the dynamic, transformative matrix of our relations with the Spirit of God, Christ in truth remains unknown—any truth will remain unknown, despite one's convictions otherwise. Only through the *logic of the spirit* does the drunken babbling become prophetically meaningful (Acts 2).

After all else that the New Testament says about the necessity of belief in Christ, how is it that Christ says that blaspheming and speaking against him will be forgiven? The only answer is that he is here exposing a necessary antecedent quality of personal openness that allows the fullness of truth within and between us. If we resist, blaspheme, speak against, or close off from this holistic antecedent encounter and conditioning by the spirit of the Other, they (or Christ) are never truly known or loved. This is the meaning of Matt 12:32. No other understanding of this passage is coherent with the balance of the scriptures, nor is its meaning accommodated within any theologies I have yet encountered, exposing their scriptural incoherence. Truth or genuine encounter with another is not a sword one possesses and wields across time, it is an emergent gift in each moment of every interaction. We might better understand it as the quality of relationship (Eph 6:17).

The two largest commentaries on Matthew that presumably researched every accessible consideration on this passage concede that Matt 12:32 is meaningless in relation to the rest of the New Testament and current theology.[4] Nevertheless, this essay's theological thesis offers a feasible and coherent understanding of this passage that accommodates and coheres with the rest of the New Testament. It also expands our theological understanding of how Christ's redemption might be active outside formal Christianity and fully expresses what Bonhoeffer theologically sensed in his letters to Bethge. Such aspirations in Bonhoeffer reveal the emergence of a greater understanding of faith beyond the current development of existing theologies, including his own. To the degree any person picks up their own cross daily, thereby allowing themselves to fully enter their relationships (love), they are already on the redemptive road, whether they know it as such or not, regardless of their existing faith confession. Those outside a confessional faith in Christ may indeed be handicapped in their redemptive development, but we can all imagine with Kierkegaard and Bonhoeffer how some outside the Christian faith might indeed be further along in their redemptive pilgrimage than many professing Christians. Though Bonhoeffer was unable in his given context and allotted time to fully articulate these ideas theologically, he lived that faith and knowledge before any coherent and final theological gestalt had emerged.

## 6. The Asymmetrical Priority of Spirit: Relational Disposition and *Analogia Spiritus*

For Bonhoeffer, a Christian pluralistic hypothesis in no way relaxes the uniqueness of Christ yet acknowledges that redemption cannot be tied emphatically to a faith confession alone. Each person develops within their own unique context, the complexity of which exceeds everyone but Christ's purview of that salvation. This is what signals the Bible's constant admonishment against judgment on such ultimate matters. How the church structures and guides its community is one thing, and there are times when communities might find it expedient to exclude specific individuals or groups. Nevertheless, emphatic judgment on the ultimate state of salvation, according to Christ, is a matter of that person's asymmetrically prior receptivity to the Spirit of God that facilitates their knowing Christ that then, and only then, brings fuller meaning to all experiences and confessions of that faith.

When the Spirit's antecedent role is given asymmetrical priority (not exclusivity) in salvation over confessional belief, it transforms our understanding of how Christ's

redemption works in the world. We continue to boldly proclaim the gospel because the full knowledge of Christ precipitates and expedites an increasingly greater fullness of life, which is much more difficult for the pagan who, at best, only indirectly and anonymously encounters Christ through a potentially genuine openness to God. Therefore, Christians can humbly release themselves from the *necessity* of a confessed belief for salvation, knowing that a person's genuine faith in Christ transcends all such confessions. As noted by Bonhoeffer earlier, this dynamic of spirit-to-Spirit encounter transcending our words and knowledge was well known by Paul: "The Spirit Himself bears witness with our spirit, . . . [making] intercession for us with groanings which cannot be uttered" (Rom 8:16,26). Any other understanding of Matthew 12:32 or leaving it to be meaningless creates scriptural incoherence. Christians should continue to call others to salvation in Christ because it is necessary for a *complete* knowing of God and greater fullness of life, but not exclusively for salvation. The shape of evangelism must emerge contextually, and therefore, we need to critically assess the shape of evangelism within each context and for each individual.

Kierkegaard suggests we proclaim the gospel without coercion or expecting specific results. Truth and belief would be better thought of as a way of becoming rather than a result.

> It requires a discipline of the spirit to honor every human being, so as not to venture directly to meddle with his God-relationship. . . . Wherever the subjective is of importance in knowledge, and where appropriation thus constitutes the crux of the matter, the process of communication is a work of art, and doubly reflected. Its very first form is precisely the subtle principle that the personalities must be held devoutly apart from one another, and not permitted to fuse or coagulate into objectivity. It is at this point that objectivity and subjectivity part from one another. (Kierkegaard 1941, p. 73)

Because of the unique transjective character of this antecedent dynamic (disposition), Kierkegaard insists we can only communicate the gospel indirectly—the listener must ultimately complete the meaning of Christ for themselves within their own meaning frame and subjective rendering spirit-to-Spirit. Only then is the Spirit of God free to attest to the listener's fullness (spirit). Therefore, only to the degree that persons fully constitute in relation to the fulness of God's Spirit will they constitute in truth. Using the metaphor of electricity, human beings must wire to God in 'parallel' rather than 'series'. Humans transmit information between humans, but all knowledge of God and knowing God in Christ, even in collective worship or the public exercise of theology, must transpire in the *analogia spiritus*—where spirit holistically interacts with Spirit and its eternal conditioning (1 Cor 2)—"for flesh and blood have not revealed this to you" (Matt 16:17). When the disciples on the road to Emmaus and the Ethiopian in the chariot experience the moment of *analogia spiritus*, Jesus and Phillip immediately disappear because the truth is now their own as they encounter a living Christ in his fullness—the authority is then internalized within them in co-conditioning action (Luke 24:13–35; Acts 8:26–40). The truth of God or identifying Christ as God cannot be communicated in 'series' from one person directly to another, technically, even from Christ (Matt 13:32). The Spirit mediates/translates the words of another into the listener's own meaning-frame, just as on that portenting day of Pentecost when all who were open to the truth heard it in their own mother tongue and meaning-frame regardless of the speaker's spoken language or meaning frame. Faith must emerge from within a person as their own faith, "for flesh and blood has not revealed this" (Matt 16:17–19). If a person's faith is not their own, they in fact do not know Christ.

In the end, there will be many whom Christ will welcome in heaven who never "confessed his name" and some who "spoke against" his name. Is it that difficult to consider that a young man's flaming atheistic anger at the church and 'Christ' because he was repeatedly raped by his Sunday school teacher might evidence his openness to the truth and injustice that unknowingly expresses an anonymous faith in Christ? We have no idea how much, if any, genuine faith such an individual ever encounters in others in their ecclesial interactions. It is simply not our call to make with any final certitude.

After centuries of missionary efforts to Japan and India, why do these cultures remain seemingly untouched by the gospel in any confessional response, yet Ireland and Korea were deeply affected to confessional response in mass within a few generations? This alone indicates how the existing relational development of a culture within its existing religious, cultural, and social dynamics conditions how Christ is understood, known, and *emerges within* that culture. Therefore, it is not the immediate conscious response to the gospel that expresses its full redemptive activity as much as each person's willingness to fully encounter and know the Other (desire for perichoretic or loving relations). If God thought it important enough to develop the nation of Israel and its faith (relationality) for thousands of years to then appropriately experience, identify, and understand Christ, then the relational development embodied within various religions and cultures certainly plays significant and diverse roles in how each experience and understand Christ (Endo 1969). Alistair MacFadyen tells us that the Christian mission is "not so much the conversion of everyone and every particular 'world' to Christianity, but their conversion to the future in which their own truth may be fulfilled, and *the universalization of the conditions of dialogue*" (MacFadyen 1993, p. 447, my emphasis—perichoresis). Understanding that our knowing Christ transcends our cultic practice allows the Church to freely love the world without diluting it with coercive agendas and inappropriate unconscious expectations. If Christ has all the time in the world (pun intended), the church should also.

Paul tells us in Romans 14 that various communities and degrees of Christian faith can exist side by side and that such paralleling communities are acceptable. The conditional development of a person or people and how Christ will genuinely emerge from within their faith experience cannot be predetermined. If Christians would trust in the Spirit of Christ's internal mediating ability to ultimately unite and translate difference (Acts 2:5–13), then they could relax their attempts to unify through forced external cultic practices and doctrinal dogma and alternatively offer such practices as supplemental and as invitation to greater intimacy with him.

## 7. The Complementarity of the Word: Co-Conditioning Action with Christ

The Spirit and Christ facilitate a reflective differentiate unity throughout creation and uniquely in humanity and all human relations. Word takes shape upon a forbearing breath and cannot be separated in their existential tension. Genuine openness and givenness to the Other necessitates that the whole of the participants enter into the whole of the relationship (Kierkegaard's notion of passion) in order to become a complete and differentiated person. Abraham and Jacob demonstrate the difference. Jacob fully enters the relationship expressing his desire to retain his ill-gotten blessing. Abraham hears the word of God and, like Job in the presence of God's transcendence, remains silent and obeys. Jacob, who maintains full autonomy, experiences God in complete perichoretic differentiation ("come of age") and passionately exposes his full desire to God (wrestling)—*concealing not the profane within him*. Jacob becomes spirit and rises to the stature of co-conditioning relation with God by expressing his *full* and profane self (the wrestling). Abraham did not. Abraham silently acquiesces to God's command without communicating his existing disappointment. But it is only upon encountering Jacob's passion and desire that God stops the presses and immediately announces that upon this kind of faith—in holistic co-condition relation to God's creating activity—will the people of God come forth. God then *completely* (not partially) changes his name and announces that upon this type of faith and holistic interaction, God will bring forth God's people, the nation of Israel—he who wrestles with God—not Moses, Abraham or David. God does not name him, 'he who is chosen by God'.

Even though Jacob wrestles against God, he is fully given to God. This is evident in Jacob's willingness to relinquish his blessing and thus bow in servitude before Esau. Nevertheless, he still expresses his full desire for the blessing with all his heart. God gives it to Jacob because of his passion, his infinite resignation before God, and also because

Christ is able to orchestrate it into the world in syncopation with all the passionate desires of others (other acts of co-conditioning faith).

Similarly, the impetuous nature of Peter (constantly speaking before he thinks), who is preeminently acknowledged by God in the New Testament, reveals that he is closer to what Bonhoeffer would call his fides directa (analogia spiritus or pre-conscious relations) rather than his fides reflexa (conscious contextualized faith). Therefore, it is not a matter of God choosing Jacob over Esau, or choosing to elevate Peter over the other disciples; these two men were simply willing to sacrifice more, and pick up more of their cross than the others. Christ is simply acknowledging this preeminently deeper action of faith that is necessary to enter the full potential of faith and know him as God, as well as genuinely knowing all others—love. God is calling humanity to a co-conditioning relationship within creation that requires *full* engagement—*analogia spiritus.* Such interactions transcend cause-and-effect relations, as all participants act in co-conditioning perichoretic unity of action through the eternal, and emerge altogether separate in time.

The story of Gideon is exemplary (Judg 6–8). God does not completely act for Gideon; God draws him into a co-conditioning relationship with Godself. The text does not reveal God telling Gideon to slay the neighboring allied leaders who were unwilling to help him (those who fearfully remained silently neutral between Gideon and the Midianites). Gideon's actions emerge as a co-conditioning effect within the *analogia spiritus* as he seemingly took it upon himself to slay those leaders who remained silently compliant with the Midianites. This parallels Bonhoeffer's minority action against Hitler versus the majority support of and silent compliance toward the Third Reich while unconsciously ignoring the suffering of others. McLaughlin notes that using Matt 25, Bonhoeffer points to "the idea that God's reward is related to righteous action" (McLaughlin 2020, p. 87), whether or not the actor is conscious that such action is christomorphic.

Nevertheless, Bonhoeffer's selfless participation with Jesus's being-for-others here expands to the greater Other, which includes the fullness of the relationship (the other, God, the world, and the *self*). This returns Jacob to the preeminence given to him by God and calls our *whole* heart—the good and the profane—into relations with God. For Christ is in the world and requires us to acknowledge the light and the darkness, for only in full exposition can the light transform the darkness (Eph 5:13). A genuine encounter with the hungry creates empathy to feed. A full encounter with the oppressed moves us to protect (for Bonhoeffer, in *fides directa*, to kill the oppressor within his pacifism). Full encounter with those in denial (not "believing all things"), move us to help them come alive to fuller truth, for Christ is the truth. To all who seek greater fullness of life, help them to acknowledge the darkness around them and in them and to "bear and endure all things" (1 Cor 13:6–7); but, let the dead bury the dead. We are called to the highest life (Rom 14) and to affirm all level of genuine openness (faith), but not closedness (the faithless), whether in the world or the church.

God calls humanity to radical openness to the Other (spirit) and yet to complete differentiated-unity of personhood into filial relations with a trinitarian God of analogous relationality. For Bonhoeffer faith necessitates, (1) being for others—openness to others and the good, *as well as* (2) reaching complete autonomy of personhood and appropriate closure to the world (McLaughlin 2020, p. 119). This reflects the irreducible nature of the Trinity as the three distinct persons in complete oneness of mutuality and action. When we passionately enter our interactions, we will ultimately act responsibly. Wholly engaging the Other will result in action on behalf of the Other (other marginalized persons, God, self, the good, truth); otherwise, we did not *fully* enter the relationship in love. If we withhold from bearing the cross of total resignation, remaining partial, we will not rise to complete co-conditioning relations with God (as friends—Jacob) but remain only good and 'faithful' servants (Abraham). For *even in Jacob's fighting God, he is fully given to God*. It is simply what committed friends and spouses do. Likewise, Bonhoeffer believes God is calling humanity into complete co-conditioning agency, *fides directa*, which is not wholly separate from *fides reflexa*, as *fides directa* emerges from the eternal *into* the interstices of

time and the contextualizing *fides reflexa*. To the degree we are willing to act ethically, we are willing to enter the full relationship—love. A maturing Christianity that properly assimilates a "world come of age" can express a non-religious interpretation of biblical concepts (McLaughlin 2020, p. 119; Bonhoeffer 2005, p. 134–45). According to McLaughlin, Bonhoeffer thinks the human ability for appropriate autonomy is dependent upon the world's coming of age (McLaughlin 2020, p. 106). The Christian "come of age" finds Christ and the sacred integrated into the interstices of all life. Bonhoeffer thought this was an ongoing process both within the social growth of the world and in theology.

> The transformation into the divine image will become ever more profound, and the image of Christ in us will continue to increase in clarity. This is a progression in us from one level of understanding to another and from one degree of clarity to another, toward an ever-increasing perfection in the form of likeness to the image of the Son of God. "And all of us, who with unveiled faces let the glory of the Lord be reflect in us, are thereby transformed into his image from glory to glory". (Bonhoeffer 2003, p. 286)

**8. Conclusions**

The inappropriate compulsion toward cultic conversion as the only initiation into the redemptive action of Christ and the Spirit is destructive to the church's relationship with the outside world; as we have seen, it does not fully square with the New Testament. There are many successful Christians and organizations that successfully mentor and disciple people and groups outside the church, and they do so effectively without proselyting those they work with. They do not force the gospel on them, but neither do they shy from boldly acknowledging it as well. Nevertheless, we can only arrest our compulsion to inappropriately convert (control and unify) if and only if we can acknowledge the theological and biblical precedent that Christ and the Spirit are working in and, to some degree, outside the church. How we ultimately identify this redemptive process is not the most important thing; what is paramount is that we acknowledge that the Scriptures are quite clear about it happening. Nevertheless, a growing understanding of how this is happening benefits the mission of the church and improves its witness to a pluralistic world.

If Matthew 12:31,32 indeed suggests that our belief in Christ becomes an idol—failing to know and encounter Jesus Christ as God—because of the bifurcation of our soul, in what Kierkegaard argues is a failure to become spirit, then our openness to the genuine expression of the Other toward affecting the entirety of ourselves signals the initiation of the redemptive process, whether it happens inside and outside the church. If the church increasingly taught this New Testament theme, expressing a more mature understanding of how Christ's redemption is active in the world while maintaining a full-bodied Chalcedonian Christology, the gospel would become quietly irresistible.

This essay offers three distinct contributions to the literature: (1) A meaningful understanding of Matt 12:31 (and Jacob) that is coherent with the rest of the Scriptures, which I have yet to find within the literature. (2) An understanding of *perichoresis* and *analogia spiritus* developed within this essay that theologically grounds a viable Christian pluralistic hypothesis. I have argued here, and elsewhere, that such an understanding leads to a more functional understanding of how a perichoretic theology might support a practical understanding of Christian inclusivism without sacrificing the uniqueness and redemptive efficacy of Christ. (3) A developing understanding of *perichoresis* and *analogia spiritus* that, through interdisciplinary theology and a metaphysical paradigm shift, open new ground for exploring answers to many age-old unresolved theological enigmas, such as divine providence and human freedom, theodicy, human origins, and of course a viable Christian pluralistic hypothesis.

**Funding:** This research received no external funding.

**Data Availability Statement:** No new data were created or analyzed in this study. Data sharing is not applicable to this article.

**Conflicts of Interest:** The author declares no conflicts of interest.

## Notes

1    The closest Bonhoeffer gets to expressing the co-conditioning dynamic within the *analogia spiritus* (and *perichoresis*) is found in (Bonhoeffer 2005, p. 330), "For just as hearing cannot be independent of [doing], so doing must not make itself independent of hearing. . . . The closest Bonhoeffer gets to expressing the co-conditioning dynamic within the *analogia spiritus* (and *perichoresis*) is found in (Bonhoeffer 2005, p. 330), "For just as hearing cannot be independent of [doing], so doing must not make itself independent of hearing. . . . Only one thing is needed—not hearing or doing as two separate things".Only one thing is needed—not hearing or doing as two separate things".

2    For a critical study of the dynamic of *perichoresis* with Bonhoeffer's paralleling theological dynamics, see (Gorsuch 2024). For a shorter presentation of a perichoretic ontology, see (Gorsuch 2022).

3    As Kierkegaard states in one of his titles, *Purity of Heart is to Will One Thing*. By this, he means (1) a person "must in truth will the good, . . . be willing to do all for it [and] . . . willing to suffer all for it," *as well as* (2) "live as an 'individual.' . . . For he who is not himself a unity is never really anything wholly and decisively" (Kierkegaard 1938, pp. 122, 184).

4    D Hagner tells us Christ's insistence that "anyone who speaks a word against the Son of Man, it will be forgiven him", is a difficult passage that does not exactly encourage optimism in the exegete (Hagner 1993, p. 347). Furthermore, W Davies and D Allison say, 'Matt 12.32 has no obvious meaning. . . . We remained stumped' (Davies and Allison 1991, p. 348), and U. Luz (2001) finds no explanation satisfactory. When Professor Hagner encountered this author's exegete of this passage in (Hagner 1993 during an exegesis class, he conceded that it indeed brought coherent meaning to the passage within the greater context of the New Testament).

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
