# Peer review of "A Christian Pluralistic Hypothesis: To Bonhoeffer and beyond—A World Christianity"

_religions, doi:10.3390/rel15010019_

Round 1

Reviewer 1 Report

Comments and Suggestions for Authors

This is an interesting paper that will be worth publishing after some further work.

There are a few clumsy phrases (what is "heavy Christology"? - line 61)

I'm unconvinced "love" is an accurate translation of "perichoresis" (line 1990

Line 443-444, with the reference to child abuse, is crass and offensive and ill suited to an academic paper. It should be removed.

The conclusion is also bombastic - the paper makes no attempt at engaging with scholarly exegesis of Matthew 12:31 but makes a dismissive assertion of a particular understanding. This needs rectifying prior to publication.

I was surprised to see no reference to Karl Rahner or Amos Yong or Van Den Toren and Tan or other scholars who have written about the Spirit and Christ being at work in the world. Clearly a short academic paper cannot cover everything or everyone, but some mention of the mass of theological work in the area of Christ and/or the Spirit at work outside the Church is needed - what is the distinctive contribution of this paper to that field?

Reviewer 2 Report

Comments and Suggestions for Authors

From a theological point of view, I have no comment to make. Instead, the conclusions part does not look like conclusions. Here I think the author should present the findings of the study, not just give a general remark.

From a technical point of view, there are some problems with the way of citing some texts, which do not respect the requirements of the journal. See line: 63-68; 79-80; 96-101; 104-109; 134-139 etc. Also, I think endnotes and not footnotes should be used.

Author Response

Thank you for your review and comments on this essay. My response to your review points are indented below each of your comments.

Reviewer 2

Thank you for your comments.

From a theological point of view, I have no comment to make. Instead, the conclusions part does not look like conclusions. Here I think the author should present the findings of the study, not just give a general remark. As noted I have completely revised the conclusion. Other reviewers suggest that as well.

From a technical point of view, there are some problems with the way of citing some texts, which do not respect the requirements of the journal. See line: 63-68; 79-80; 96-101; 104-109; 134-139 etc. Also, I think endnotes and not footnotes should be used.  This was a formatting error evidently from transferring my essay from my submitted Word doc to the MDPI formatting. I adjusted the Word MDPI copy they sent, so hopefully that will be fixed.

Reviewer 3 Report

Comments and Suggestions for Authors

Overall, I find this research very interesting. The manuscriptA Christian Pluralistic Hypothesis: To Bonhoeffer and Beyond—A World Christianity”  appears to be an underexplored area. I sincerely congratulate you.

I acknowledge that I found pleasure in reading this text. However, I would like to provide some constructive suggestions that you might want to consider:

1. Including references to the works of C. Lewis and S. Weil in the list of references would enhance the depth of your paper, especially since you have referenced excerpts from their works at the beginning (Lines 19-28).

2. The presence of an introduction, or at least an indication of it, could contribute to the clarity of the paper. Consider placing a simple Introduction in front of the title "Bonhoeffer's Lived Faith and Unconscious Christianity."

3. There seems to be a missing page reference in Line 40: "McLaughlin 2020:155–156." Ensure that the page numbers are included.

4. Lines 89-90 contain separated text. Before submitting the work to the editors, it would be advisable to carefully review the text to address any technical improvements.

5. In Line 217, the work "Santorum communion" is missing letter c. Please verify and add the necessary letter.

6. In the footnotes 2, it appears that the letters "Jacob and the Night of Faith: Analogia Spiritus" are larger than others. Ensure consistent formatting for uniformity.

7. The parallel drawn in Lines 270-273 between household chores seems somewhat out of place and unusual. Consider reevaluating its relevance to the overall work.

8. Personal testimonies, such as those in Lines 486-488 where you speak about your own period of pre-Christian atheism and contempt for Christianity, may not be suitable for a scientific work. Consider revising or removing such elements.

9. Line 601, "Let us passionately move forward in Christ's eternal care and the Spirit's freedom," seems more fitting for a sermon than a scientific work. Scientific writing typically demands a more formal and objective expression. Consider reformulating for a more academic tone.
